# A Novel Computer-Aided Detection/Diagnosis System for Detection and Classification of Polyps in Colonoscopy

**DOI:** 10.3390/diagnostics13020170

**Published:** 2023-01-04

**Authors:** Chia-Pei Tang, Hong-Yi Chang, Wei-Chun Wang, Wei-Xuan Hu

**Affiliations:** 1Division of Gastroenterology, Department of Internal Medicine, Dalin Tzu Chi Hospital, Buddhist Tzu Chi Medical Foundation, Chiayi 622401, Taiwan; 2School of Medicine, Tzu Chi University, Hualien City 970374, Taiwan; 3Department of Management Information System, National Chiayi University, Chiayi City 60054, Taiwan

**Keywords:** colon polyp detection, generative adversarial network (GAN), object detection, data computer augmentation, image deblurring

## Abstract

Using a deep learning algorithm in the development of a computer-aided system for colon polyp detection is effective in reducing the miss rate. This study aimed to develop a system for colon polyp detection and classification. We used a data augmentation technique and conditional GAN to generate polyp images for YOLO training to improve the polyp detection ability. After testing the model five times, a model with 300 GANs (GAN 300) achieved the highest average precision (AP) of 54.60% for SSA and 75.41% for TA. These results were better than those of the data augmentation method, which showed AP of 53.56% for SSA and 72.55% for TA. The AP, mAP, and IoU for the 300 GAN model for the HP were 80.97%, 70.07%, and 57.24%, and the data increased in comparison with the data augmentation technique by 76.98%, 67.70%, and 55.26%, respectively. We also used Gaussian blurring to simulate the blurred images during colonoscopy and then applied DeblurGAN-v2 to deblur the images. Further, we trained the dataset using YOLO to classify polyps. After using DeblurGAN-v2, the mAP increased from 25.64% to 30.74%. This method effectively improved the accuracy of polyp detection and classification.

## 1. Introduction

In recent years, the incidence of colorectal cancer (CRC) has increased because of refined and high-fat diets. The survival rate of CRC patients is positively correlated with early detection and treatment. The American Association for Cancer Research states that CRC is one of the most prevalent types of cancer [1]. Colon polyps are precursors of CRC [2]. However, the rate of misdetection of colon polyps during colonoscopy is as high as 26%, as reported in a recent meta-analysis [3]. The 5-year survival rate of early stage CRC exceeds 90% with a well-established screening program [4]. The American Society for Gastrointestinal Endoscopy has proposed “resect and discard” and “diagnose and leave” strategies, suggesting that non-neoplastic polyps can be left unresected [5,6,7]. This strategy is based on real-time polyp classification, which is cost-effective and saves time for endoscopists. Methods have been developed to enhance the early detection and classification of colon polyps [8,9].

Considerable progress has been made in the field of deep learning and in the development of computing hardware. Artificial intelligence (AI)-based systems used to assist physicians in clinical diagnosis have flourished [10,11]. Abnormal images are common issues in medical-related AI model training, and they are scarce. Among colon polyps, sessile serrated adenoma (SSA) is the rarest and has a morphology similar to that of a hyperplastic polyp (HP). An adequate amount of data or images is required for a workable machine-learning model [12]. An imbalance in the quantity and variability of images or datasets leads to a poor classification model [13,14,15,16,17]. This study proposes methods for solving the shortage of datasets used in model training to improve the accuracy of colon polyp detection.

This study primarily aimed to use the collected polyp dataset to develop a high-accuracy object detection model for two types of neoplastic colon polyps (tubular adenoma (TA) and SSA) and a non-neoplastic polyp (HP). These three types of polyps were annotated by the trained model to aid the endoscopist in removing neoplastic polyps during colonoscopy.

Two types of deep learning studies have been conducted on colon polyps: image classification studies [18,19,20,21,22] used to determine the presence and type of a polyp in an image and object detection studies [23,24,25] used to detect the polyp location. Image classification studies aim to comply with the “resect and discard” and “diagnose and leave” strategies in clinical practice. Object detection studies are mainstream because endoscopists need to identify the location of a polyp before removing it.

In current published datasets and studies, polyps have seldom been classified. Colon polyps are generally classified as neoplastic or nonneoplastic. Neoplastic polyps include TA and SSA, whereas nonneoplastic polyps include HPs. Recognizing the type of polyp helps the endoscopist decide whether to resect it. In particular, SSA progresses into CRC within a short duration and needs to be removed at the earliest. The shortage of polyp datasets has been discussed in previous studies [26,27,28]. Shin et al. [29] used a data augmentation method to increase the number of datasets, including rotation, scaling, cropping, blurring, and brightness changes. The augmented dataset had different angles for the same image and had certain limitations. Blurring, brightening, and darkening of the images reduce the accuracy of the model. A flat SSA is a rare neoplastic polyp of the colon. This study proposes using a generative adversarial network (GAN) framework to generate an SSA database [30], and combining DeblurGAN-v2 and YOLOv5 [31] to improve the sensitivity of colon polyp detection and classification. After labeling the features of the SSA ground truth, we used a condition to overlay them on two other polyp images to generate a mimic image with the same background and SSA features. This technique increases the quantity and diversity of SSA databases.

Constant searches for polyps and movement of the colonoscope during the procedure cause polyp blurriness, which reduces the model’s polyp detection ability. Studies have focused on endoscopic motion blurriness restoration [32,33,34,35]. The model is typically trained using high-definition images, and the detection rate decreases for blurred polyps. Another adverse effect is an increase in the number of false positives (FP), which distracts the endoscopist and hampers the adoption of the system.

The following are the contributions of our study to enhanced real-time polyp detection and classification.

For a comprehensive and balanced model of colon polyp classification, the shortage of an SSA is an issue. We supplemented it with the GAN method and achieved superior results compared with the data augmentation technique.The blurriness caused by the endoscopy movement decreases polyp detection accuracy. We used a motion-blurred restoration model to deblur the images and improve the detection.Our GAN-generated database on the state-of-the-art YOLOv5 algorithm performs outstanding compared with other algorithms.The add-on effect of the GAN method increases the accuracy of polyp detection and classification compared with other studies.

## 2. Related Studies

Fonollà et al. [18] implemented a computer-aided diagnostic system for colorectal polyps using a CNN-based image-classification technique. Their dataset included three enhancement techniques: white light (WL), blue light (BLI), and linked color imaging (LCI). EfficienNetB4 was used as the CNN architecture. After training the model, 60 new colorectal polyp images were provided to the computer-aided system and six endoscopists for evaluation. The accuracy, specificity, and sensitivity of the six endoscopists were 81.7%, 94.1%, and 61.1%, respectively. For Fonollà’s model, the results were 95.0%, 93.3%, and 95.6%, respectively. The specificity of the model was slightly lower than that of endoscopists. The other parameters were significantly higher than those of endoscopists. In terms of sensitivity, the model resulted in better identification of potential neoplastic polyps.

Qadir et al. [26] used the Mask R-CNN architecture for implementation and applied the same datasets CVC-ClinicDB and ETIS-Larib from MICCAI 2015 as Brandao et al. [36]. They used the data augmentation method proposed by Shin [13], and Mask R-CNN models with Resnet50, Resnet101, and Inception Resnet as the backbone were learned through transfer learning using pre-trained parameters from Microsoft’s COCO. The results revealed a recall of 64.42% and a precision of 70.23% for Resnet50. For Resnet101, a recall of 72.59% and a precision of 80.0% were shown. In addition, the recall of Inception ResNet was 64.9%, and the precision was 77.6%. Resnet101 showed the best polyp-detection performance.

Li et al. [28] implemented a computer-aided diagnostic system using a convolutional neural network (CNN) image classification technique. Acquiring a polyp dataset from a collaborating hospital is expensive; thus, the authors obtained a free dataset collected from public websites such as YouTube, VideoGIE, and Vimeo. A total of 23 videos ranging from 1 to 41 min long were obtained, and the extracted images were augmented through panning, flipping, and blurring. The final dataset consisted of 15,270 images of polyps, including 2310 SSA images and 20,625 background images. The AlexNet transfer learning technique was used with pre-trained parameters from ImageNet on Caffe. The final AlexNet model achieved a sensitivity of 73% and a specificity of 96%. This sensitivity was lower than the 90% reported in other studies. However, its specificity was higher than that reported in other studies (80%).

Shin et al. [29] used the CVC-ClinicDB dataset from MICCAI 2015. The considerable diversity in polyps made the public database unrealistic in terms of model training. Similar polyps and backgrounds appear repeatedly, affecting the detection rate of the object detection model. Therefore, we used a conditional GAN approach to enhance the dataset. They first outlined the polyps in white and marked all their ground truths with a white background. The remaining images are marked in black. The contour was treated as a condition, and the original image was treated as the correct answer for conditional GAN training. The generator had a normal five-layer U-Net architecture and replaced the original convolution with a dilated convolution, which obtained a wider range of features and reduced the size of the final feature map. After training, the GAN was provided with a contour map to generate the corresponding polyp. They pasted a white background of the ground truth polyp on the colon image without a polyp to simulate a real polyp in the colon mucosa. This fake contour map was included in the trained GAN to generate the corresponding fake polyp. In total, 372 fake images were generated. Fake and real images were trained using a Faster R-CNN model. The precision was 67.2% without GAN and 79% with GAN, whereas the recall was 61% without GAN and 75% with GAN. Both parameters revealed that the model trained using GAN achieved better results.

In addition to the above-mentioned image classification techniques used to determine the presence of polyps, studies using object detection techniques for searching the ground truth of polyps have also been conducted. Fan et al. [37] used Res2Net as the base architecture. Unlike Res2Net, which integrates all layers, the authors discarded the information of the low-level layers (the first two layers of the network) and used a parallel partial decoder to aggregate only the high-level layers (the last three layers of the network). Their study used reverse attention to identify the areas of a polyp by removing foreground objects. The mean dice indicator was 0.899. Brandao et al. [36] used a fully convolutional network (FCN) architecture, replacing the fully connected layers of the network with inverse convolutional layers, restoring the image size by upsampling the feature map, and marking the location of the polyps. The backbone networks AlexNet, GoogLeNet, and VGG were used. CVC-ClinicDB was used as the training dataset, and ETIS-Larib was used as the test dataset. Both are from the 2015 MICCAI competition.When applying FCN-AlexNet, a recall of 63.78% and precision of 44.08% were attained; for FCN-GoogLeNet, a recall of 65.76% and precision of 41.85% were achieved; and for FCN-VGG, a recall of 86.31% and precision of 73.61% were achieved. FCN-VGG exhibited the best polyp detection performance.

Debesh et al. [38] applied an image-segmentation method to a medical database. They used the CVC-ClinicDB dataset from MICCAI, 2015. There are two types of image-segmentation techniques: FCN and U-Net. The DoubleU-Net architecture was developed based on U-Net. The first U-Net encoder used the VGG19. The single block of the decoder was a 2 × 2 bilinear upsampling layer, and the corresponding VGG-19 feature map was then concatenated, followed by a convolution layer and a batch normalization layer. The inputs of the second U-Net were the inputs and outputs of the first U-Net. The encoder block had two convolution layers and one batch normalization layer, which reduced the internal co-variant. Although the structure of the decoder block was the same as that of the first U-Net, the decoder of the second U-Net used both VGG-19 feature maps and its own feature map from the encoder. The final output was the result of an output concatenated from the two versions of the U-Net. DoubleU-Net was used to recognize the CVC-Clinic data and obtained a recall of 0.85, which was higher than that of the traditional U-Net at 0.79. In addition, the precision of 0.96 was higher than that of U-Net at 0.93, and the mIoU at 0.86 was higher than that of U-Net at 0.79.

The real-time application of a trained object detection model is crucial. Many object detection models have achieved a high detection rate with a low frame rate. YOLOv5 [27] has high detection and frame rates and is a one-stage object-detection model suitable for the main algorithm described in this study.

## 3. Materials and Methods

The colonoscopy images were based on colonoscopies performed with high-definition colonoscopes (CF-H290I; Olympus, Tokyo, Japan) at Dalin Tzu Chi Hospital, a teaching hospital in Taiwan, from July 2021 to January 2022, with the approval of the Institutional Review Board (B11003002). The polyp detection experiments were divided into two parts: the training of a conditional GAN and YOLO. Three models were trained to generate the TA, SSA, and HP images. SSA images are scarce in the real world, whereas those of the other two polyp types are common. The GAN model was trained using all three polyp types under the same conditions. In Section 3.10, the detailed amount and type of experimental data are explained. The performance indicators are explained in Section 3.11.

### 3.1. Modifying the Aspect Ratios of GAN Training Data

The original training dataset had three resolutions of 640 × 480, 720 × 480, and 1980 × 1080 pixels. The GAN model training process consumed a large amount of GPU resources, and all images were scaled to the same pixel resolution of 512 × 512 pixels.

### 3.2. GAN Training Data Labeling

“LabelMe” was used to label the GAN training data, allowing the user to click on multiple dots next to the polyps to form an irregularly shaped part of the image. It generated a JSON file that recorded the locations of all dots. Each point coordinate in the JSON file was converted into an image, resulting in the example shown in Figure 1. The red part represents the labeled polyp, and the black block represents the background.

### 3.3. Finding Contours Using the Canny Algorithm

The Canny edge algorithm was used to detect the contours of the polyp image, and its input was restricted to grayscale images. The RBG three-channel colon polyp image was converted to a grayscale image. Canny edge detection is a fast process that is easily affected by noise and overmarked contours. Gaussian blurring was therefore implemented to eliminate noise in an image prior to Canny edge detection. The kernel of the Gaussian filter was set to (9 × 9), and the “color σ” (standard deviation of the color space) was set to zero. The larger the “color σ, the greater the weight of the point farther from the center. There were two threshold values for the Canny edge detection, and the level of detail of the contour map differed depending on the threshold value. After the tests were conducted, a small threshold value of 60 and a large threshold value of 100 were found to be the most effective for marking the contours of the polyp images (Figure 2).

### 3.4. Overlaying a Contour Map on a Ground Truth Map

After generating a contour map, the polyp contours were pasted onto the ground-truth map marked in Section B. The Canny contour map is a single-channel greyscale image. The Canny contour map was first converted into a three-channel RGB color map, which was pasted on the red ground truth map (Figure 3).

### 3.5. Overlaying the Ground Truth Map onto the Original Image

We trained the conditional GAN to generate images of different types of polyps with the same background. To maintain the stability of the generated background, the ground truth part of the original image was changed into an image from the previous step (Figure 4).

### 3.6. Conditional GAN Architecture

This study applied a conditional GAN framework, which differs from a general GAN. The input was not a random number but a conditional image allowing the GAN to generate an image corresponding to the condition. The condition was the conditional graph generated in Section E, and the target image we intended to generate was the target image mentioned later. The target image is the original true image. A fake image generated by the generator is called a generated image. First, the conditional image was sent to the generator to generate the corresponding image. The target image, conditional image, and generated image were then sent to the discriminator to determine whether they were true or fake, and gradients were applied. The GAN generator used was a DoubleU-Net proposed by Jha et al. [38], which performed better than Unet and its extensions in the image segmentation of medical images. We discarded the original five-layer U-Net in the original study on a conditional GAN, replaced it with DoubleU-Net, and modified several parameters to fit the output. Figure 5 shows the architecture of the DoubleU-Net model.

Encoder 1 is a VGG19, followed by an ASPP, which helps extract high-resolution feature maps, followed by decoder block 1 (1-1 through 1-4). A decoder block contains 2 × 2 bilinear up-sampling, which can double the size of the image, concatenate the corresponding VGG feature maps, and then connect to the convolution and batch normalization layers. The original DoubleU-Net output 1 generates a single-channel image with a black background and a white area for the polyp. Our model generated a three-channel polyp image. We simulated the FCN approach and directly output a three-channel image by removing the sigmoid layer. The first input (input of U-net 1) and output 1 were multiplied by the input of the second U-Net and entered into encoder block 2 (2-1 through 2-4). Each Encoder block has two convolution layers followed by a batch normalization layer and is then connected to the squeeze_exite_block to enhance the quality of the feature maps. Squeeze_excite_block contains GlobalAveragePooling2D and two dense layers and is then connected to an ASPP and Decoder block 2 (2-1 through 2-4). In the last step, an additional deconvolution (Conv2DTranspose) is added to convert the result of the concatenation into a new feature map. The result of concatenation was transformed to be the same size as the original image.

The discriminator network was ResNet50. The architecture was imitated from PatchGAN, which turned the final output into a 30 × 30 × 1 tensor. The determined input was a true image based on these 900 results. There were two types of inputs for the discriminator. The first input is the conditional and target images generated by Section E (which should be judged as true), and the second is the conditional and generated images (which should be judged as false). When applying the gradient process, the conditional images are concatenated with a target or generated image. A Conv2DTranspose layer was applied to match the input size of ResNet50 and input into ResNet50. ResNet50 uses the pre-trained ImageNet weight and removes the top layer. The feature map slowly decreased in size, and thus inverse convolution, ZeroPadding, convolution, BatchNormalization, LeakyReLU, ZeroPadding, and finally, a convolution layer was added. The output shapes were set as (30, 30, 1). These 900 parameters were used to determine the truth or falsity of images. The loss function is based on the manner in which the PatchGAN is expressed. The original discriminator output of the binary value (real and fake) is converted into a 30 × 30 × 1 matrix, which is then used to discriminate between real and fake images with a matrix shape of 30 × 30 × 1 made up of all zeros and 1s. A GAN has two networks, a generator, and a discriminator, each of which has its own loss function. Figure 6 shows the loss calculation process for the generator. The loss function of the generator consists of two main losses, one of which is the binary cross-entropy, which calculates the (30, 30, 1) output of the discriminator. The other has the same shaped matrix, where all values are 1s. The other loss was the L1 mean absolute error (MAE) loss, which is the average absolute value of the generated images minus the target images. With these two losses, the total loss was calculated, and the algorithm was the binary cross-entropy loss + (LAMBDA × L1 loss), where LAMBDA was changeable and set to 100 to apply the gradients.

Figure 7 shows the loss calculation process of the discriminator, which received two sets of images: the first was a conditional image and a generated image, and the second was a conditional image and a target image. Binary cross-entropy was calculated for both groups. As the difference between groups, the first was calculated with the same shape (30, 30, 1), and zero was applied for all values, whereas the values of the other group were all 1 s. This method was used because the discriminator classified the generated image as a negative sample. The loss with (30, 30, 1) was calculated as all 0 s, and the discriminator was used to classify the target image as a positive sample. The loss with (30, 30, 1) was calculated using all 1s. Finally, the discriminator loss algorithm was binary cross-entropy (0 s) + binary cross-entropy (1 s) and gradients were applied.

### 3.7. Comparison of GAN Output

The original conditional GAN (Pix2Pix) was compared with the proposed GAN. The input size of the two GANs was 512 × 512 pixels. The differences between the three types of polyp images generated by the two models and the original images were compared. The variables are Peak Signal to Noise Ratio (PSNR) and Structural Similarity Index (SSIM). These two variables are introduced below: In addition to comparing these two variables, a visualized comparison of the two GAN-generated images is also attached.

#### 3.7.1. Peak Signal-to-Noise Ratio (PSNR)

The PSNR expresses the pixel difference between two images and is often used to measure the signal reconstruction quality when compressing the images. The PSNR is defined as a simple MSE error in Equation (1):(1)$$MSE=\frac{1}{mn}\sum_{i=0}^{m-1}\sum_{j=0}^{n-1}{\left[I\left(i,j\right)-K\left(i,j\right)\right]}^{2}$$
where *mn* represents the size of the image (length × width), *I* and *K* are two different images, and (*i*, *j*) is the value of the image at pixel position (*i*, *j*). This equation represents the sum of the squares of the disparity between images *I* and *K* at each pixel divided by the image size. In addition, the peak signal-to-noise ratio is defined by Equation (2).
(2)$$PSNR=10\cdot \log_{10}\left(\frac{MAX_{I}{}^{2}}{MSE}\right)=20\cdot \log_{10}\left(\frac{MAX_{I}}{\sqrt{MSE}}\right)$$
where *MAX_I_* is the maximum value. If each point is represented by 8 bits, the total value is 255. The PSNR algorithm is the square of *MAX_I_* divided by the MSE and multiplied by 10. If the pixel gap between the two images is smaller, the value of the MSE is smaller, and the PSNR of these two images will become larger, which means that the two images are similar.

#### 3.7.2. Structural Similarity Index Measure (SSIM)

The SSIM is an index used to measure the similarity between two images. Compared with the PSNR, the SSIM is more in line with the human judgment of image quality. Given two signals *x* and *y*, the SSIM is defined by Equation (3).
(3)$$SSIM\left(x,y\right)={\left[l\left(x,y\right)\right]}^{\alpha }{\left[c\left(x,y\right)\right]}^{\beta }{\left[s\left(x,y\right)\right]}^{\gamma }\;l\left(x,y\right)=\frac{2\mu_{x}\mu_{y}+C_{1}}{\mu_{x}{}^{2}+\mu_{y}{}^{2}+C_{1}},\;c\left(x,y\right)=\frac{2\sigma_{x}\sigma_{y}+C_{2}}{\sigma_{x}{}^{2}+\sigma_{y}{}^{2}+C_{2}},\;s\left(x,y\right)=\frac{\sigma_{xy}+C_{3}}{\sigma_{x}\sigma_{y}+C_{3}}$$
where *l*(*x*, *y*) compares the brightness of *x* and *y*, *c*(*x*, *y*) compares the contrast of *x* and *y*, and *s*(*x, y*) compares the structure of *x* and *y*. In addition, *α*, *β*, and *γ* are the parameters used to adjust the relative importance of *l*(*x*, *y*), *c*(*x*, *y*), and *s*(*x*, *y*), all of which should be greater than zero. In addition, *μ_x_* and *μ_y_* represent the mean of *x* and *y*, respectively; *σ_x_* and *σ_y_* represents the standard deviations of *x* and *y*; and *σ_xy_* represents the covariance of *x* and *y*. Moreover, *C_1_*, *C_2_*, and *C_3_* are constants. If there are two identical images, that is *μ_x_* = *μ_y_* and *σ_x_* = *σ_y_*, then SSIM = 1. The SSIM is close to 1, and the similarity between the two images is greater. To simplify its use, the SSIM is generally set to *α* = *β* = *γ* = 1 and *C_3_
*= *C_2_*/2. The simplified equation is as follows:(4)$$SSIM\left(x,y\right)=\frac{\left(2\mu_{x}\mu_{y}+C_{1}\right)\left(2\sigma_{xy}+C_{2}\right)}{\left(\mu_{x}{}^{2}+\mu_{y}{}^{2}+C_{1}\right)\left(\sigma_{x}{}^{2}+\sigma_{y}{}^{2}+C_{2}\right)}$$

### 3.8. YOLO Data Labeling

The labeling tool of YOLO is LabelImg, which pulls a box around the target object (bounding box) and assigns a corresponding number or class name to the bounding box, resulting in a record file labeled Pascal VOC. An experienced endoscopist labeled the image data. The GAN-generated images and the original images share Pascal VOC because the polyp positions on both images are the same. The difference lies in the change in the class name. For example, the class name of the original image converted from a TA image to an SSA image is changed to *SSA*. However, the position of the polyp remained unchanged, and the bounding box of Pascal VOC was maintained.

### 3.9. YOLO Data Augmentation

For data augmentation, the following methods and rules were used to generate the two augmented images:(a)A random rotation of the image within plus or minus 45 degrees.(b)Random horizontal and vertical shifts of 20% were made toward the left and right sides of the image, respectively.(c)Random zoom in or out of the image (80–120% of the original image).(d)We ensured that the ground truth remained in the image when the above actions were conducted.

### 3.10. Training YOLO

The parameters of YOLO were as follows. The image length and width were set to 608. The batch size was set to 16 because of the GPU memory shortage. The number of training epochs was set to 6000, and the learning rate was set to 0.001. All the other training parameters were the same as those used in the original YOLO. We collected the original TA, SSA, and HP images reviewed by our certified and experienced pathologists. Then, we extracted the images from the pre-recorded videos during colonoscopy. The SSA, TA, and HP groups each had 700 images and were divided into training, validation, and testing groups. There were 600, 50, and 50 images for training, validation, and testing, respectively. All three polyp-type images were randomly selected and were not repeated. The “Ori” label was used to represent the 600 SSA, TA, and HP training images. The label “Aug” represents the augmented images, and “N GAN” indicates the images generated using the original data and N GAN data. Additionally, 150, 300, 450, and 600 GANs were compared. Here, 150 GAN indicates 150 images using 50 images for each of the three classes. Several models were trained, as shown in Table 1, and were divided into the original dataset (Ori), a dataset trained with the original data plus data augmentation (Ori + Aug), and a dataset trained with the original data and GAN data. To determine whether the number of images used in the GAN affected the accuracy, all GAN data were separated, and a total number of 600 GANs were applied.

### 3.11. Comparison of Model Metrics

The true positive (TP), false positive (FP), false negative (FN), precision, recall, average precision (AP) of the three categories, mean average precision (mAP), and IoU (Figure 8) were compared. Precision and recall were evaluated by drawing a curve-treated precision as the y-axis and recall as the x-axis of the detected object. The curve is labeled as a PR curve, and the area under the curve indicates the AP of the class. The mAP is the average of all AP values. To avoid the result from being affected by a single model, we tested each model five times and averaged the results.
(5)$$precision=\frac{TP}{TP+FP}$$
(6)$$recall=\frac{TP}{TP+FN}$$

### 3.12. Gaussian Blurring of the Test Video

Gaussian blurring was used to adjust the frames of the video to simulate the blurred images from movement during colonoscopy. The average blurring algorithm was not used because it produces unnatural images. The average blurring aims to turn each point in the picture into the average of all points next to it. If the kernel was set to (3 × 3), the center point was the average of eight surrounding points, whereas if the kernel was (5 × 5), it was an average of 24 surrounding points in two circles, and so on.

The weight of the kernel setting is positively related to the distance from the point to the center and not the average of all points, regardless of the distance. We used Gaussian blur to blur the video, which calculated the normal distribution of the two-dimensional space and assigned a weight to each point in the kernel (including the center point), with a total weight of 1. The color σ (standard deviation of the color space) was set to effectively simulate movement during colonoscopy. This value indicates that the points farther from the center have more weight, which produces a more blurred image. In this study, the kernel setting was (15 × 15), and the color σ was set to 10.

After blurring the video, DeblurGAN-v2 deblurring was applied. The pretrained InceptionResNet-v2 architecture of DeblurGAN-v2 was used, which achieved a better deblurring effect than other generator networks, such as MobileNet. The training data were obtained from the GoPro dataset containing 3214 high-definition photographs captured by a GoPro Hero 4 camera.

### 3.13. Evaluating the Deblurring Effect

Ten-fold cross-validation was applied to another YOLO model, which classified the images into two classes (neoplastic and non-neoplastic). The test images were the test sets in ten-fold cross-validation, and the test data were Gaussian filtered and DeblurGAN-v2. The detection process is illustrated in Figure 9. The metrics that were compared were TP, FP, FN, precision, recall, F1, SSA_AP, TA_AP, mAP, and IoU.

## 4. Results

### 4.1. Comparison of GAN-Generated Images

The original Pix2Pix architecture (512 pixels × 512 pixels) was compared with the proposed conditional GAN composed of DoubleU-Net and ResNet50. The SSIM and PSNR values of the three classes (HP, TA, and SSA) were compared. These metrics indicate the similarity between and structure of the generated and original images. The PSNR was rounded to the first decimal place, and the SSIM was rounded to the second decimal place. The quality of the generated images was compared visually. SSIM and PSNR are compared in Table 2, Table 3 and Table 4.

The SSIM approximates 1, and a larger PSNR indicates a better performance. DoubleU-Net and ResNet50 demonstrated superior performances for all three categories. For the PSNR of HP, DoubleU-Net and ResNet50 were 1.3 higher than those of Pix2Pix, and the SSIM was 0.1 higher. For the PSNR of SSA, DoubleU-Net and ResNet50 were 0.3 higher than those of Pix2Pix, and the SSIM was 0.4 higher. For the PSNR of TA, DoubleU-Net and ResNet50 were 0.6 higher than those of Pix2Pix, and the SSIM was 0.2 higher. For the PSNR metric, HP achieved the best performance, followed by TA, and SSA. For the SSIM metric, HP performed the best, followed by SSA and TA. This study speculated that HP generated better images, probably because HP had fewer feature details and GAN learned limited features. The GAN performed best for the HP images. In contrast, SSA and TA had complicated features, and GAN achieved suboptimal performance. Figure 10, Figure 11 and Figure 12 show the images generated by Pix2Pix (left) and GAN (right).

The output of Pix2Pix on the background was almost the same as that of DoubleU-Net and ResNet50 GAN (Figure 12). Our GAN had fewer artificial artifacts than Pix2Pix, and its overall appearance was better.

### 4.2. Comparison of YOLO Using Different Datasets

Six YOLO models were trained. Here, Ori indicates the original data, Ori + Aug is the original data with data augmentation, and NGAN represents the original data with N GAN-generated images. In total, 150, 300, 450, and 600 GANs were compared. The data are the average results obtained after five training sessions.

The results were similar to those of the larger GAN datasets (Table 5). For the best performance, with the highest TP, precision, and recall, and the lowest FP and FN, was 150 GAN. In addition, 450 GAN and 600 GAN achieved a higher FP, presumably because of the misjudgment of YOLO when using a large amount of GAN data. The size of the GAN dataset should be adjusted appropriately.

Here, 300 GAN achieved the highest AP for both SSA and TA, whereas 150 GAN achieved the highest AP for HP (Table 6). All GAN models had a higher mAP than that of the Ori model. The mAP of both 450 and 600 GAN was lower than that of Ori + Aug. The mAP values of the 150 GAN and 300 GAN were higher than those of Ori + Aug. These findings are consistent with the results presented in Table 5. We speculate that the excessive GAN data may have misled the model. In conclusion, the GAN data with 1/12 of the training data achieved the best performance.

### 4.3. Comparison of Gaussian Blur and DeblurGAN-v2

The Gaussian blurred images and those generated by DeblurGAN-v2 were compared visually. Figure 13 shows the Gaussian blurred image, Figure 14 shows the DeblurGAN-v2 deblurred image, and Figure 15 shows the zoomed-in images.

The deblurring effect of DeblurGAN-v2 is shown with white dots (Figure 14). The white dots and polyps were blurred, and the borders were indistinguishable (Figure 13). After the deblurring effect was achieved by DeblurGAN-v2, the white dots became distinguishable from the background.

### 4.4. Comparison of YOLO Using DeblurGAN-v2

Here, “G” indicates the results of Gaussian blur, and “D” shows the results of DeblurGAN-v2. The tested models were classified into two classes (neoplastic and nonneoplastic). The following tables show the average results of the ten-fold cross-validation test presented in Table 7. TP, FP, and FN included decimal points, and precision, recall, and F1 were rounded to the second decimal place. The performance indicators are represented by real numbers in Table 5 and Table 7, as we used a 10-fold cross validation mechanism in the experiment.

Table 7 lists the YOLO results of Gaussian blur and DeblurGAN-v2. The TP and FN of DeblurGAN-v2 achieved better performance. The FP was suboptimal. The precision, recall, and F1 were improved. Table 8 shows the AP, mAP, and IoU of the Gaussian blur and DeblurGAN-v2.

Table 8 shows the AP, mAP, and IoU of Gaussian blur and DeblurGAN-v2. All indicators improved with DeblurGAN-v2, that is, SSA_AP by 1.84%, TA_AP by 8.36%, mAP by 5.1%, and IoU by 1.51%. Using DeblurGAN-v2 to deblur blurred images improves the detection rate of the object detection model.

### 4.5. Performance Comparison with Other Object Detection and Classification Models

To demonstrate the performance of our proposed computer-aided detection/diagnosis system in comparison with other famous detection and classification models, we trained YoloV3, YoloV4, YoloR, and YoloX on the same polyp dataset that we used to train the YoloV5 architecture. The results presented in Table 9 and Table 10 reveal that the YoloV5 architecture outperforms other modes in terms of mAP and precision. YoloV5 remains the best choice for fine-tuning polyp datasets for better detection and classification results. The experimental results show that the detection and classification of polyps are not very effective. The main reason for this is that the polyps are too small in colonoscopy examination and the characteristics of different polyps are very similar, so it is not easy to distinguish the types. Table 11 shows the analysis of the error range for each value.

### 4.6. Model Optimization

To improve the classification accuracy of the YOLOv5 model, this study aims to modify the model’s parameters. The first step is to increase the number of training epochs (which is set to 300 by default) and the input image size (which is set to 640 by default). According to the results shown in Table 12, increasing the number of training epochs beyond 350 does not significantly improve accuracy and may cause overfitting. In addition, increasing the input image size does not improve the classification accuracy for colorectal polyps and may even decrease accuracy due to reduced image resolution.

To further improve the YOLOv5 model, we experimented with different model architectures. We evaluated five architectures: YOLOv5n, YOLOv5s, YOLOv5m, YOLOv5l, and YOLOv5x. The number of layers and parameters for each model is shown in Table 13. The inference speed for all of the models was around 10 milliseconds, with YOLOv5s being the fastest at approximately 9.8 milliseconds and YOLOv5x being the slowest at 11.2 milliseconds due to its larger number of layers and more complex parameters. In terms of accuracy, the results in Table 14 show that YOLOv5l had the best overall performance, with a precision of 80.18%, the highest among the five models. Considering the balance between inference speed and accuracy, the YOLOv5l is the most suitable model architecture for classifying polyps.

## 5. Discussion

Colorectal cancer (CRC) is a preventable disease. Early detection of colon polyps prevents their progression to CRC. Although a computer-aided detection system was developed to aid endoscopists, misdetection of polyps still occurs. The American Society for Gastrointestinal Endoscopy proposed a “resect and discard” and “diagnose and leave” strategy, and suggested leaving a non-neoplastic polyp in situ. Therefore, reliable computer-aided detection and diagnostic systems are essential. However, a large image database is required to train a high-accuracy computer-aided diagnosis and classification model. SSA is a rare type of neoplastic polyp. We used a conditional GAN to generate SSA images to supplement the database. A contour map combined with a mask filter for the polyps and background was used as the condition for a conditional GAN. A conditional GAN with DoubleU-Net used as a generator and ResNet50 used as a discriminator was applied to overlay different classes of polyps on an original polyp image and to generate a diverse polyp database. In the YOLO results, the 300 GAN model showed the highest AP of 54.60% and 75.41% for SSA and TA, respectively. This result was better than that of the data augmentation method, for which the AP of SSA and TA was 53.56% and 72.55%, respectively. In addition, the 300 GAN model achieved the highest AP, mAP, and IoU of the HP at 80.97%, 70.07%, and 57.24%, respectively. These results were better than those of the data augmentation method, which showed values of 76.98%, 67.70%, and 55.26%, respectively. It is important to note that the GAN data achieved better results, and the amount of GAN data should be appropriately adjusted.

Blurred images that occur during a colonoscopy procedure are another issue. A blurred image misleads the object detection model, and an endoscopist might, therefore, miss a polyp. The DeblurGAN-v2 was used to address this issue. We used a Gaussian filter to simulate a blurred image taken during colonoscopy and then applied DeblurGAN-v2 to deblur the image. The results indicate that, although the use of DeblurGAN-v2 might increase FP, better TP, FN, precision, recall, F1, mAP, and IoU are achieved. The mAP increased by 5.1%, from 25.64% to 30.74%. All results indicate that using the proposed GAN and DeblurGAN-v2 improves colon polyp detection capability during colonoscopy.

Our study had several limitations. First, the images were obtained from a single hospital. Datasets from different locations could increase the accuracy of the model. Second, the images used to establish the model were selected to avoid duplicate polyps. Increasing the number of single-polyp images is challenging because it requires a large number of patients. Third, our model for detection and classification works effectively in real-time. Implementation in a real clinical scenario requires further investigation. The application of the model in a real-time colonoscope and conducting a clinical study will be the subject of a future study.

## 6. Conclusions

Our data showed that the polyp detection accuracy after implementing the proposed GAN method to generate rare polyp images was better than that obtained with the commonly used augmentation technique. The results indicated that an optimal sample number was crucial for achieving a better GAN outcome compared with non-GAN training. Colon polyp detection and classification sensitivity were improved with GAN and DeblurGAN-v2 combined with the YOLOv5 method.

## Figures and Tables

**Figure 1 diagnostics-13-00170-f001:**
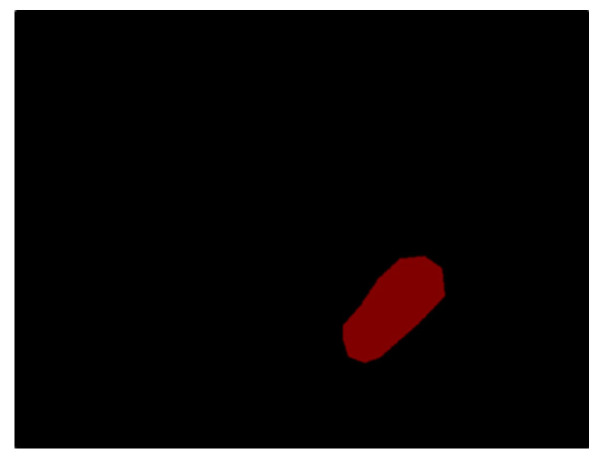
“LabelMe” labeled polyp and background.

**Figure 2 diagnostics-13-00170-f002:**
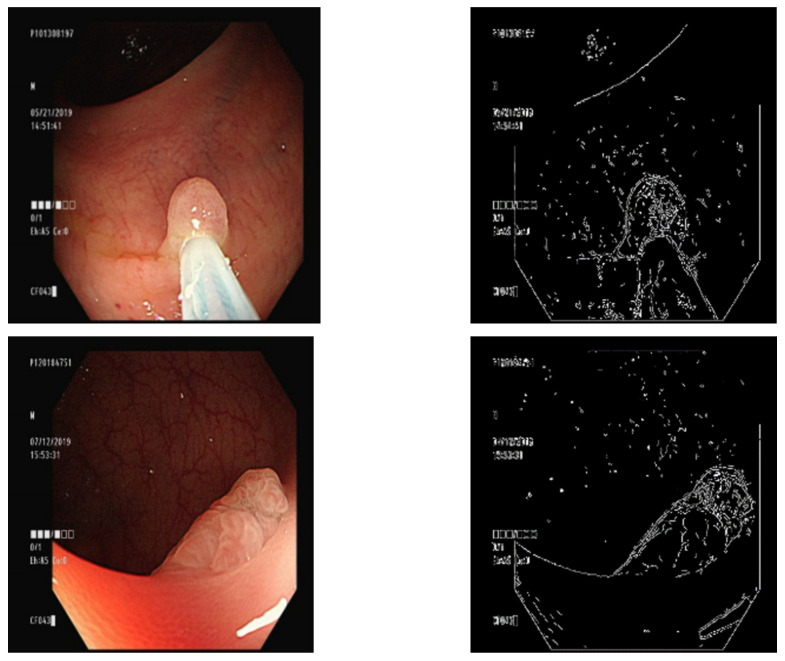
Canny edge marking the contour of a detected polyp.

**Figure 3 diagnostics-13-00170-f003:**
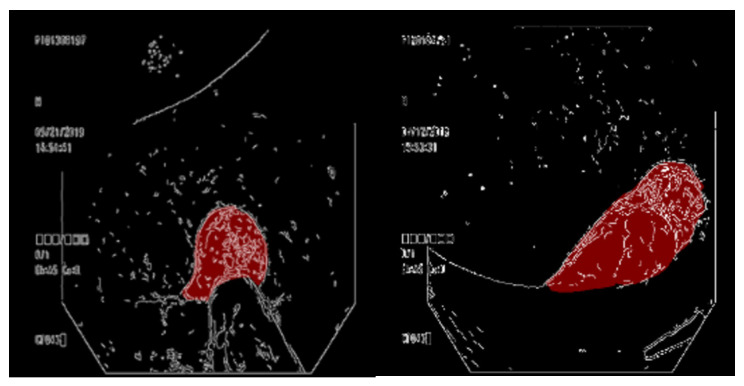
Contour map placed over the ground truth map.

**Figure 4 diagnostics-13-00170-f004:**
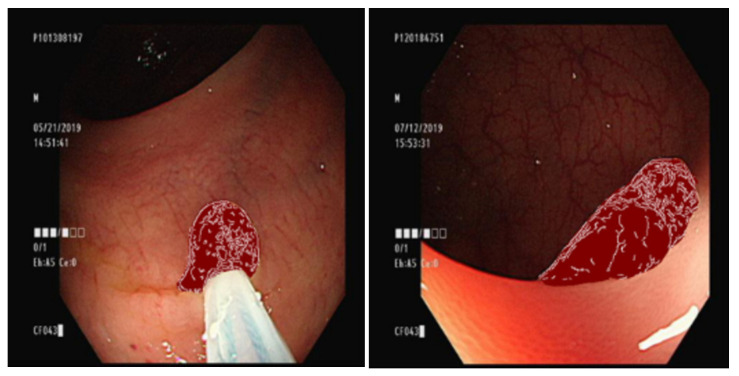
The condition of the GAN.

**Figure 5 diagnostics-13-00170-f005:**
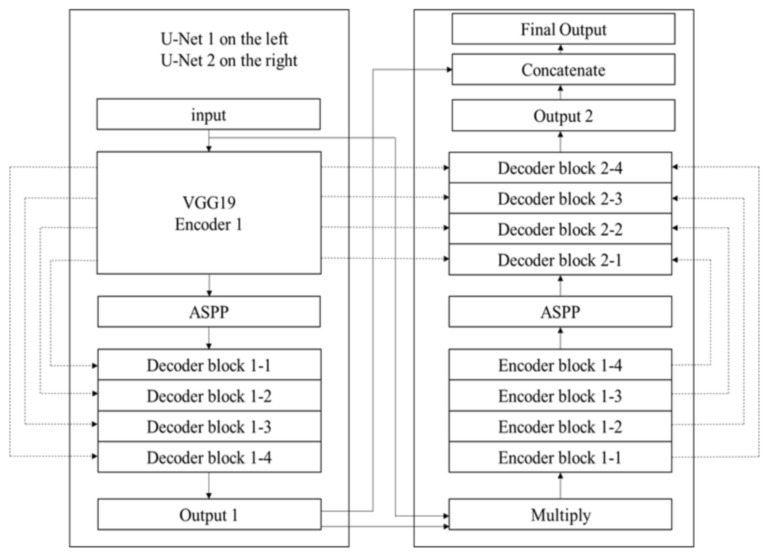
The architecture of the DoubleU-Net model.

**Figure 6 diagnostics-13-00170-f006:**
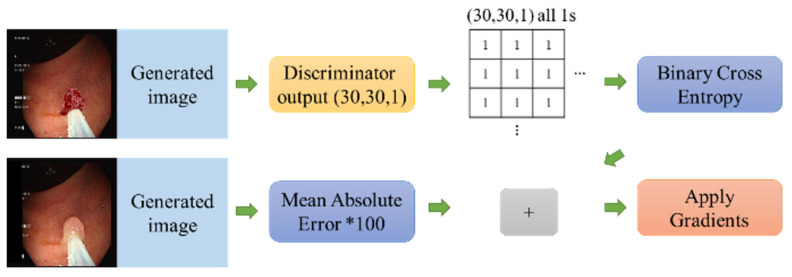
The loss calculation process of the generator.

**Figure 7 diagnostics-13-00170-f007:**
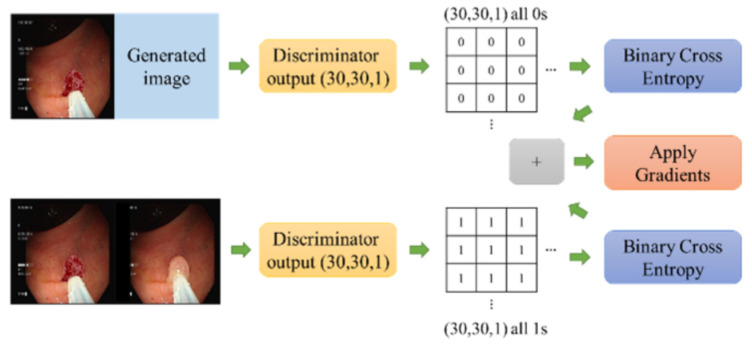
The loss calculation process of the discriminator.

**Figure 8 diagnostics-13-00170-f008:**
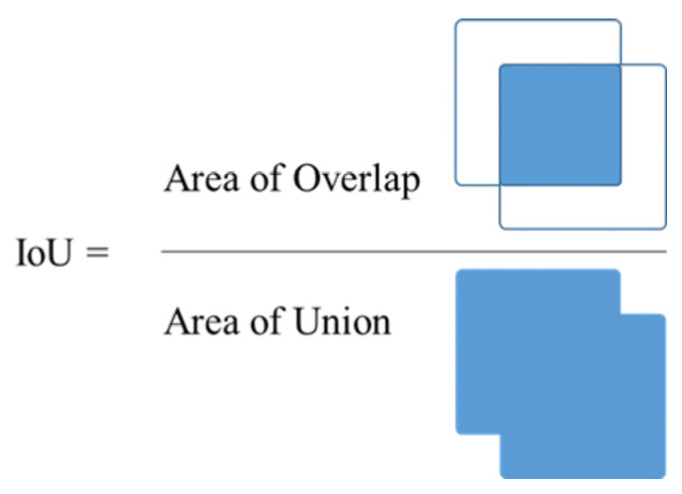
Intersection over Union (IoU).

**Figure 9 diagnostics-13-00170-f009:**
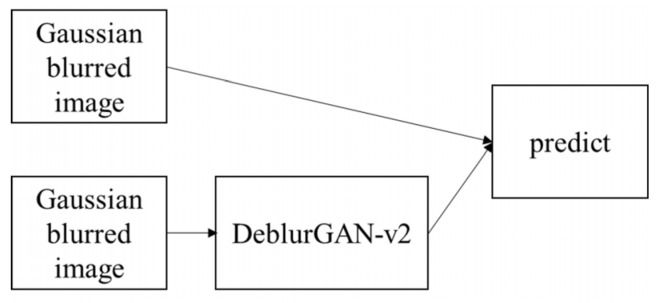
Deblurred detection process.

**Figure 10 diagnostics-13-00170-f010:**
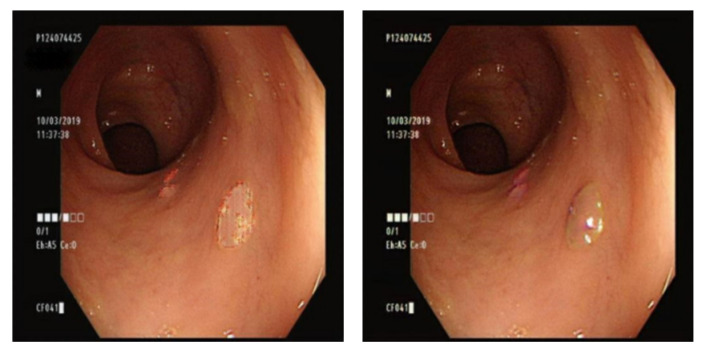
Pix2Pix and GAN HP images.

**Figure 11 diagnostics-13-00170-f011:**
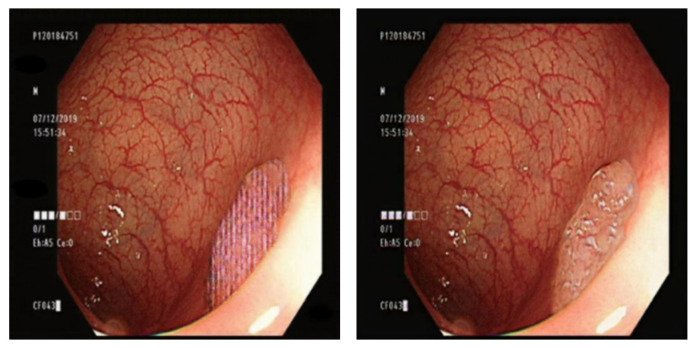
Pix2Pix and GAN SSA images.

**Figure 12 diagnostics-13-00170-f012:**
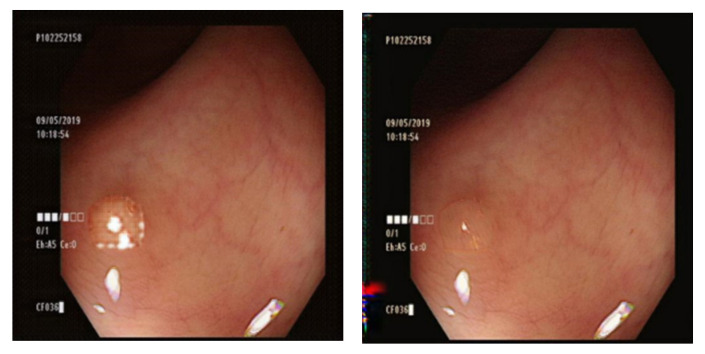
Pix2Pix and GAN TA images.

**Figure 13 diagnostics-13-00170-f013:**
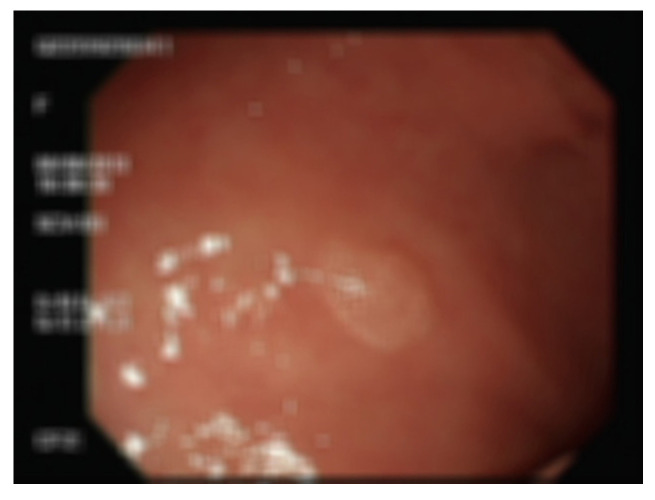
Gaussian blurred image.

**Figure 14 diagnostics-13-00170-f014:**
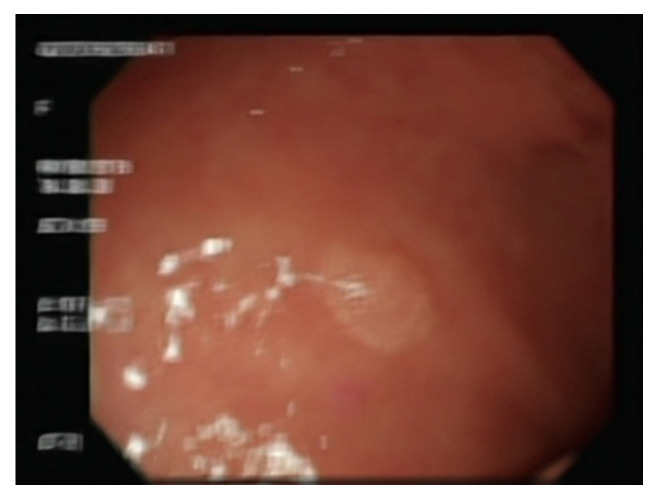
DeblurGAN-v2 deblurred image.

**Figure 15 diagnostics-13-00170-f015:**
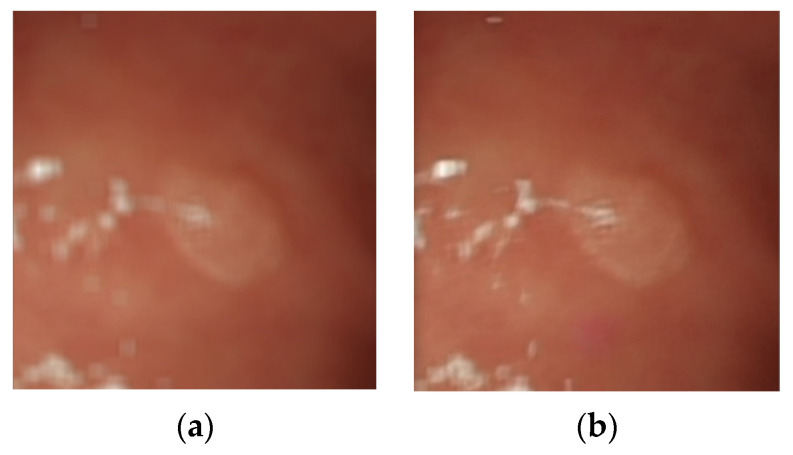
Zoomed-in image. (**a**) Gaussian blur; (**b**) DeblurGAN-v2.

**Table 1 diagnostics-13-00170-t001:** Training image data distribution.

Ori
Ori + Aug
Ori + 150 GAN
Ori + 300 GAN
Ori + 450 GAN
Ori + 600 GAN

**Table 2 diagnostics-13-00170-t002:** Comparison of SSIM and PSNR for HP.

	Avg of PSNR	Avg of SSIM
Pix2Pix	24.4	0.92
Ours	25.7	0.93

**Table 3 diagnostics-13-00170-t003:** Comparison of SSIM and PSNR for SSA.

	Avg of PSNR	Avg of SSIM
Pix2Pix	21.4	0.86
Ours	21.7	0.90

**Table 4 diagnostics-13-00170-t004:** Comparison of SSIM and PSNR for TA.

	Avg of PSNR	Avg of SSIM
Pix2Pix	22.8	0.80
Ours	23.4	0.82

**Table 5 diagnostics-13-00170-t005:** Comparison of TP, FP, FN, and precision recall for a GAN.

	TP	FP	FN	Precision	Recall
Ori	102.0	49.6	68.0	0.67	0.60
Ori + Aug	105.4	47.8	64.6	0.69	0.62
150 GAN	111.4	47.6	58.6	0.70	0.66
300 GAN	110.8	49.8	59.2	0.69	0.65
450 GAN	106.0	54.8	64.0	0.66	0.62
600 GAN	109.6	53.8	60.4	0.67	0.65

**Table 6 diagnostics-13-00170-t006:** Comparison of AP, mAP, and IoU (GAN).

	SSA	TA	HP	mAP	IoU
Ori	52.03%	72.42%	75.72%	66.71%	54.83%
Ori + Aug	53.56%	72.55%	76.98%	67.70%	55.26%
150 GAN	54.08%	75.17%	80.97%	70.07%	57.24%
300 GAN	54.60%	75.41%	77.36%	69.12%	55.53%
450 GAN	51.56%	71.62%	77.66%	66.95%	53.44%
600 GAN	50.59%	72.23%	78.96%	67.26%	54.25%

**Table 7 diagnostics-13-00170-t007:** Comparison of TP, FP, FN, precision, and recall (Deblur).

	TP	FP	FN	Precision	Recall	F1
G	39.1	60.8	119.8	0.39	0.25	0.30
D	43.3	63.6	115.6	0.40	0.27	0.33

**Table 8 diagnostics-13-00170-t008:** Comparison of AP, mAP, and IoU (Deblur).

	SSA	TA	mAP	IoU
G	26.63%	24.65%	25.64%	28.95%
D	28.47%	33.01%	30.74%	30.46%

**Table 9 diagnostics-13-00170-t009:** Comparison of mAP and IoU for different models.

	SSA	TA	HP	mAP	IoU
Yolov3	33.68%	51.67%	45.74%	43.70%	44.22%
Yolov4	44.72%	73.03%	71.32%	61.72%	52.77%
Yolov5	61.01%	77.97%	78.06%	72.37%	45.80%
YoloR	58.28%	77.18%	76.62%	70.67%	45.83%
YoloX	57.12%	75.79%	76.73%	69.88%	46.12%

**Table 10 diagnostics-13-00170-t010:** Comparison of TP, FP, FN, precision, and recall for different models.

	TP	FP	FN	Precision	Recall
Yolov3	58.20	36.1	125.7	0.62	0.32
Yolov4	111.1	55.3	72.80	0.67	0.61
Yolov5	138.5	41.2	45.40	0.76	0.67
YoloR	129.5	64.4	54.40	0.67	0.70
YoloX	155.2	67.4	28.70	0.70	0.84

**Table 11 diagnostics-13-00170-t011:** The analysis of the error range for each value.

	Range of Accuracy	Range of Error	Standard Deviation
Yolov3	35.8% (min.)~46.8% (max.)	43.70% ± 1.00%	3.27%
Yolov4	59.7% (min.)~66.6% (max.)	61.72% ± 1.52%	2.25%
Yolov5	68.5% (min.)~75.8% (max.)	72.37% ± 0.60%	2.86%
YoloR	64.0% (min.)~73.8% (max.)	70.67% ± 5.01%	3.13%
YoloX	66.4% (min.)~75.5% (max.)	69.88% ± 3.07%	3.27%

**Table 12 diagnostics-13-00170-t012:** The comparison of mAP for different Epochs.

	Img-Size	640	1280
Epochs			
200		65.47%	52.52%
300		67.06%	59.95%
The highest value		73.47% (Epoch = 352)	66.89% (Epoch = 360)
400		71.71%	61.84%
500		70.29%	64.87%
600		70.25%	62.96%
700		70.45%	62.39%
800		70.65%	63.84%
900		71.35%	62.60%

**Table 13 diagnostics-13-00170-t013:** Parameters for different model architectures.

Model Name	Layers	Parameters	Inference Speed
Yolov5n	213	1,779,460	10.3 ms
Yolov5s	213	7,050,580	9.8 ms
Yolov5m	213	1,779,460	10.2 ms
Yolov5l	367	46,183,668	9.9 ms
Yolov5x	444	86,267,620	11.2 ms

**Table 14 diagnostics-13-00170-t014:** The comparison of accuracy for different model architectures.

	Yolov5n	Yolov5s	Yolov5m	Yolov5l	Yolov5x
Precision	0.7857	0.7452	0.7806	0.8018	0.7559
Recall	0.6874	0.6453	0.6720	0.6765	0.6874

## Data Availability

Not applicable.

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
