# Peer review of "A Novel Computer-Aided Detection/Diagnosis System for Detection and Classification of Polyps in Colonoscopy"

_diagnostics, 2023, doi:10.3390/diagnostics13020170_

Round 1

Reviewer 1 Report

This paper uses Gaussian blurring to simulate the blurred images during a colonoscopy and then applies DeblurGAN-v2 to deblur the pictures of the polyp. The authors train the dataset using YOLO to classify the polyps. After using DeblurGAN-v2, the mAP increased from 25.64% to 30.74%. This method improves the accuracy of polyp detection and classification effectively. Overall, this paper is well written and the research topic is interesting. However, in order to make it more attractive, the authors are advised to modify this paper as listed below.

1.     For better readability, the authors may expand the abbreviations at every first occurrence.

2.     Some minor typos and grammatical errors (although these errors do not stop the reader to understand the work of the authors) should be avoided. The reviewer strongly suggests that the authors should go through the entire manuscript many times to remove all of them.

3.    Please ensure that all references in the Reference list are cited in the text and vice versa.

4.   Several typo errors are throughout this paper. The authors should carefully check their paper.

5.    The reference format of the paper is inconsistent. Please follow reference instructions of Diagnostics for improvement.

6.     The resolution of the images in the paper could be enhanced.

Reviewer 2 Report

1.      The authors did not describe different types of TA, SSA, HP and the number of images. It is difficult to verify the effectiveness of their system.

2.      The image generated by using the generative adversarial network is too similar to the original image. It is hard to apply the system to real application.

3.      SSA and HP are sometimes difficult to be differentiated even by pathologists. It is needed to describe how to diagnose of SSA in this study.

4. The study model achieved average precision of 54.6% for SSA and 75.41 % for TA. The precision rate is too low for clinical application, suggesting the system has some defects.

Reviewer 3 Report

This manuscript regarding the colorectal cancer incidence used a 10 deep learning algorithm in the development of a computer-aided system for colon polyp detection 11 and was effective . The aim of this study was to develop a system for colon polyp 12 detection and herein classification was well presented. I would like to sugesst to discuss range of error in each values prresented.

Thank you, Sincerely 

Round 2

Reviewer 2 Report

1. The title of the third paragraph should be changed to Materials and Methods, and clearly state the amount and type of original experimental data.

2. The performance indicators such as TP, FP, FN and recall in 3.1 are not clearly defined. Are they integer values? This will make readers unable to understand why there are real numbers in Table 5 and 7?

3. The accuracy of the overall system performance is not high. It is recommended to adjust the system architecture and deep learning parameters instead of just using the typical architecture for experiments.
